# Effect of Botulinum Toxin Injections in the Treatment of Spasticity of Different Etiologies: An Umbrella Review

**DOI:** 10.3390/ph17030310

**Published:** 2024-02-28

**Authors:** Iris Otero-Luis, Arturo Martinez-Rodrigo, Iván Cavero-Redondo, Nerea Moreno-Herráiz, Irene Martínez-García, Alicia Saz-Lara

**Affiliations:** 1Health and Social Research Center, Universidad de Castilla-La Mancha, 16071 Cuenca, Spain; iris.otero@uclm.es (I.O.-L.); irene.mgarcia@uclm.es (I.M.-G.); alicia.delsaz@uclm.es (A.S.-L.); 2Departamento de Sistemas Informáticos (DSI), Facultad de Comunicación de Cuenca, University of Castilla-La Mancha, 16071 Cuenca, Spain; arturo.martinez@uclm.es; 3Facultad de Ciencias de la Salud, Universidad Autonoma de Chile, Talca 3460000, Chile

**Keywords:** general population, botulinum toxin injections, spasticity, etiologies, umbrella review

## Abstract

Background: Spasticity is a very common neurological sequelae that significantly impacts the quality of life of patients, affecting more than 12 million people worldwide. Botulinum toxin is considered a reversible treatment for spasticity, but due to the large amount of available evidence, synthesis seems necessary. Therefore, we conducted an overview of existing systematic reviews and meta-analyses to evaluate the effect of botulinum toxin injections in the treatment of spasticity of different etiologies. Methods: A systematic search of different databases, including Pubmed, Scopus, the Cochrane Library, and Web of Science, was performed from inception to February 2024. Standardized mean differences (SMDs) and their respective 95% confidence intervals (CIs) were calculated to assess the effect of botulinum toxin compared to that of the control treatment using the Modified Ashworth Scale (MAS). All the statistical analyses were performed using STATA 15 software. Results: 28 studies were included in the umbrella review. The effect of botulinum toxin injections on spasticity, as measured by the MAS, was significantly lower in all but three studies, although these studies also supported the intervention. The SMDs reported by the meta-analyses ranged from −0.98 to −0.01. Conclusion: Botulinum toxin injections were effective at treating spasticity of different etiologies, as indicated by the measurements on the MAS. This implies an improvement in muscle tone and, consequently, in the patient’s mobility and quality of life.

## 1. Introduction

Spasticity is a highly prevalent neurological sequelae that significantly affects quality of life. It manifests as an intrinsic increase in muscle resistance related to the speed of the tonic reflex during passive stretching of a limb in individuals with upper motor neuron syndrome [1] (Figure 1). Spasticity leads to complications such as pain, a distorted joint position, postural and hygiene difficulties, joint contractures, and permanent deformities [2]. It is common in patients with central nervous system disorders such as stroke, multiple sclerosis, spinal cord injuries [3], traumatic brain injuries, and cerebral palsy (CP) [1]. Approximately 12 million people worldwide are estimated to suffer from spasticity [4].

There are two scales used specifically to assess spasticity. The first is the Modified Ashworth Scale (MAS) [1,3], an ordinal scale that measures the intensity of muscle tone on a scale of 0 to 4 [3]. On this scale, 0 represents “no increase in tone,” while 4 indicates that the limb is “stiff in flexion or extension” [5] (Appendix A). The second is the Tardieu Scale or the Modified Tardieu Scale [1,3], which is also an ordinal scale that assesses the intensity of resistance to muscle stretch [3]. According to the Modified Tardieu Scale, each muscle group is rated for a specific stretching speed according to two parameters. The first is the quality of the muscle reaction, where 0 indicates no resistance to movement and 4 indicates that the occurrence of clonus lasted more than 10 s. The second parameter is the angle of muscle reaction, measured at the minimum stretching position of the muscle for all joints except the hip, where it is assessed in relation to the anatomical position at rest [5] (Appendix A).

The treatment of spasticity requires a multidisciplinary team, which includes medical specialists, occupational therapists, physiotherapists, nurses, and nutritionists, among others, to achieve the best results and improve patient quality of life [1]. There is evidence of the efficacy of different treatments for the improvement in spasticity, such as physiotherapy (stretching, use of orthoses, cryotherapy, heat, etc.), extracorporeal shock waves, oral medication, injections (such as botulinum toxin), intrathecal baclofen pumps, and surgical interventions [3]. In particular, botulinum toxin is considered a reversible treatment for spasticity and has been used for more than 30 years in patients with CP [6]. In poststroke patients, studies have shown an improvement in muscle tone, resulting in safe and effective therapy [7]. The different types of botulinum toxin injections used include Botox, Dysporter, Neurobloc, and Xenomin [6]. The administration dose depends on the patient’s weight, the number of muscles to be treated, the severity of spasticity, the size of the muscle, and the type of toxin [6,7].

Several systematic reviews and meta-analyses have shown that botulinum toxin injections are effective at reducing spasticity [8,9,10,11,12,13,14,15,16,17], with a peak effect occurring at 5 weeks [17] and decreasing efficacy at 12 weeks postintervention [10]. It is considered a safe therapy and is likely to improve the quality of life of poststroke patients [15,16]. Furthermore, it is an effective intervention for reducing spasticity in children with spasticity [18], particularly in children with CP [19]. Since no study has synthesized all the existing information on the effect of botulinum toxin injections, an umbrella review is needed to synthesize this information and evaluate the effect of the treatment on the reduction in spasticity. Therefore, we conducted an overview of existing systematic reviews and meta-analyses to assess the effect of botulinum toxin injections in the treatment of spasticity of different etiologies.

## 2. Methods

This umbrella review was conducted following the Preferred Reporting Items for Systematic Reviews and Meta-Analyses (PRISMA) guidelines [20] and the Cochrane Collaboration Handbook [21]. This study was registered in PROSPERO (Registration Number: 502078).

To conduct this umbrella review, a search strategy has been implemented using keywords and Boolean operators, following inclusion and exclusion criteria to identify all available reviews on the treatment of spasticity with botulinum toxin injections. Once the studies were selected, two tables were created to classify them, and their quality was assessed. Finally, data synthesis was performed.

### 2.1. Search Strategy

A search was conducted in the Pubmed, Scopus, Cochrane, and Web of Science databases from their inception to February 2024. The selected keywords were combined using Boolean operators (AND, OR) following the population, intervention, comparator, outcomes (PICO) strategy to identify studies assessing the effect of botulinum toxin in the treatment of spasticity (Appendix A). The search strategy used was as follows: ((“general population”) OR (“children”) OR (“adults”)) AND ((“botox”) OR (“botulinum toxin”)) AND ((“spasticity”) OR (“cerebral palsy”) OR (“spastic paraplegia)) AND ((“systematic review”) OR (“meta-analysis”) OR (“network meta-analysis”)). In addition, reference lists of the retrieved systematic reviews and meta-analyses were checked for additional studies.

### 2.2. Inclusion and Exclusion Criteria

The inclusion criteria were as follows: (i) population: general population (children and adults); (ii) intervention: botulinum toxin injections; (iii) outcome: spasticity assessed with the MAS and/or Modified Tardieu Scale; (iv) study design: systematic reviews and/or meta-analyses; and (v) no language restriction. The exclusion criteria were as follows: (i) articles combining botulinum toxin injections with other treatments; and (ii) articles evaluating outcomes other than efficacy and effectiveness in improving spasticity.

### 2.3. Data Extraction

Two ad hoc tables were created, one for systematic reviews and one for meta-analyses, where the data from the selected studies were included as follows: (1) reference (first author and year of publication); (2) study design (only for meta-analyses); (3) number of included studies; (4) type of population; (5) age range; (6) intervention; (7) comparator; (8) length of intervention (weeks); (9) spasticity assessment scale; (10) effect; and (11) assessment of study quality using the AMSTAR 2 scale (Table 1 and Table 2).

### 2.4. Methodological Quality Assessment

The quality of the selected studies was assessed using the AMSTAR 2 tool [39]. This tool critically evaluates the risk of bias in systematic reviews and consists of 16 different domains that assess relevant methodological aspects, each of which is answered “yes”, “no”, “cannot answer”, or “partial yes”. The overall quality of the studies was rated as high when there were no deficiencies or only one non-critical deficiency; moderate when there was more than one non-critical deficiency; low when there was a critical deficiency with or without non-critical deficiencies; and critically low with more than one critical deficiency with or without non-critical deficiencies.

Two researchers (I.O-l and A.S-L) independently conducted the study selection, data extraction, and assessment of the methodological quality of the included studies. Disagreements were resolved through consensus or by a third reviewer (I.C-R.).

### 2.5. Grading the Quality of Evidence

We used the Grading of Recommendations, Assessment, Development, and Evaluation (GRADE) tool [40] to assess the quality of evidence and provide recommendations. Each outcome had a high, moderate, low, or very low evidence score, depending on the study design, risk of bias, inconsistency, indirect evidence, imprecision, and publication bias.

### 2.6. Data Synthesis

The DerSimonian and Laird random-effects method [41] was used to calculate pooled estimates of standardized mean differences (SMDs) and their respective 95% confidence intervals (95% CIs) to assess the effect of botulinum toxin injections compared to that of a control group using the MAS and they are displayed in a forest plot.

Heterogeneity was examined using the I2 statistic [42], which ranges from 0% to 100%. Based on the I2 values, heterogeneity was considered not important (0–30%), moderate (30–60%), substantial (60–75%), or considerable (75–100%). The corresponding *p* values were also considered.

A forest plot was generated using Stata SE software, version 15 (StataCorp, College Station, TX, USA).

## 3. Results

### 3.1. Baseline Characteristics

The flowchart was created to summarize how the data extraction process was carried out in this manuscript.

Of the 519 manuscripts collected from different databases, 110 records were selected for full-text review after screening by title. Finally, 28 manuscripts were included in the umbrella review (Figure 2).

Two tables were created, one for systematic reviews (Table 1) and one for meta-analyses (Table 2), showing the characteristics of the studies included in this umbrella review. Of the total number of included studies, 12 were systematic reviews [22,23,24,25,26,27,28,29,30,31,32,33] and 16 were meta-analyses [8,10,11,12,13,14,15,16,17,18,19,34,35,36,37,38]. The studies included in this umbrella review were published between 2001 and 2023. The range of participants included in the studies was between 1 and 468. The study participants were children and adults with spasticity caused by CP or stroke aged between 8 months and 92 years. The duration of intervention ranged from 0 to 48 months. Almost all the included studies used the MAS [8,10,11,12,13,14,15,16,17,18,19,22,23,24,26,27,28,29,30,31,32,33,34,35,36,38,42], five used the Tardieu Scale [11,18,24,31,32], and five used the Modified Tardieu Scale [13,19,25,27,33].

In summary, two tables were created, one for systematic reviews and one for meta-analysis, which synthesize the data obtained from the records included in this study.

### 3.2. Methodological Quality Assessment and GRADE

To assess the quality of the studies, it is necessary to use a specific scale for systematic reviews and meta-analysis.

The methodological quality of the studies was assessed using the AMSTAR 2 tool. Of the total included studies, 35.71% were rated “high”, 50.00% were rated “moderate”, 7.14% were rated “low”, and 7.14% were rated “critically low” (Appendix A).

When the GRADE was evaluated, 92.86 of the pairwise comparison studies were rated as low and 7.14 as very low (Appendix A).

In summary, most of the studies have achieved moderate quality in the AMSTAR 2 tool and most of the studies were rated as low in the GRADE tool.

### 3.3. Data Synthesis

A forest plot of the included meta-analyses has been conducted to assess the effectiveness of botulinum toxin injections on spasticity.

Figure 3 shows the forest plot of the meta-analyses included in the umbrella review assessing the effectiveness of botulinum toxin injections compared to the control group. The SMD reported by the meta-analyses ranged from −0.98 to −0.01. All the studies except three showed a significant reduction in spasticity measured with the MAS (18,38,39), which, although not significant, was in favor of the intervention (standardized mean difference (SMD): −0.04; 95% CI: −0.14, 0.05; SMD: −0.27; 95% CI: −0.80, 0.26; and SMD: −0.1; 95% CI: −0.3, 0.1). Overall heterogeneity in the forest plot was considerable (94.9%). Therefore, there is a significant reduction in spasticity and a reduction in muscle tone according to the MAS.

In summary, in all studies, the results were favourable to the intervention, showing significant results in 78.57% of the studies.

### 3.4. Effectiveness of Botulinum Toxin Injection

The impact of botulinum toxin injections in clinical practice needs to be evaluated, as shown below. Our results indicate that botulinum toxin decreases spasticity assessed by the MAS. Studies suggest that in clinical practice, this therapy reduces spasticity, decreasing muscle tone, resulting in a reduction in the MAS [15,19]. This reduction occurs both in poststroke patients [15] and in children with CP [19].

In summary, both in the results obtained and in clinical practice, a reduction in spasticity is shown.

## 4. Discussion

### 4.1. Main Findings

This umbrella review aimed to analyze the effect of botulinum toxin injections in the treatment of spasticity of different etiologies using the MAS. This study provided significant evidence on the effectiveness of botulinum toxin injections in reducing muscle tone, as measured by the MAS, as indicated by a decrease in spasticity.

### 4.2. Interpretation

According to the studies reviewed, botulinum toxin injections can improve lower limb spasticity in children with CP [19] and can be used in combination with other therapies, such as physiotherapy [29,38] or physical activity [27]. One study supported the efficacy of botulinum toxin injections in children under two years of age with CP and highlighted the safety of its use [33]. Regarding patients with poststroke spasticity, the authors reported that botulinum toxin injections reduce spasticity in both the upper and lower limbs, suggesting that botulinum toxin is a safe treatment option [11,12,14,15,22,26]. However, according to one study, the effects of the treatment decreased 12 weeks after application [10].

Botulinum toxin, when injected at high concentrations, strongly affects the management of spasticity by blocking the release of the neurotransmitter acetylcholine from the neuromuscular junction to various muscle groups, resulting in what is known as “chemical denervation” [3]. Neuromuscular blockade affects both intrafusal and extrafusal muscle fibers. The decrease in activity of intrafusal muscle fibers (reduction in afferent input) peaks at 2 weeks and gradually decreases by 12 weeks post-injection. This blockade reduces the flow of muscle spindles to spinal stretch reflex circuits, reducing spasticity [2] (Figure 4). Botulinum toxin doses are usually adjusted according to factors such as the severity of spasticity, number of muscles affected, age, previous response to botulinum toxin treatment, and the use of adjuvant therapy. The development of antibodies against botulinum toxin proteins can lead to therapeutic failure. To avoid this, increasing the dose of botulinum toxin, avoiding short intervals between injections, and using different botulinum toxin serotypes are suggested [7]. Based on the studies reviewed, this therapy is generally well tolerated and safe, although botulinum toxin injections can cause fatigue, tiredness, pain, skin rashes, flu-like symptoms, worsening of spasms, weakness, convulsions, and incoordination [12,26]. Compared to other therapies, such as intrathecal baclofen, the incidence of treatment-related adverse effects is low [27].

Botulinum toxin is the most widespread treatment for spasticity [43], although there are other effective treatments, such as intrathecal baclofen pumps; surgical procedures, such as selective dorsal rhizotomy; physiotherapy, such as extracorporeal shockwave therapy; and oral medication [3]. Botulinum toxin is a very versatile treatment option because it can be combined with other agents, such as extracorporeal shockwave therapy, physiotherapy and rehabilitation, and splints and casts [2].

A study in poststroke patients, in which different doses were analyzed, reports that regardless of the dose and type of botulinum toxin used, there was a reduction in the MAS, with the maximum reduction occurring between weeks 4 and 6 post-injection [16]. One of the reviewed studies reports that there was no difference in spasticity reduction between the application of 100 U of botulinum toxin combined with short-term electrical stimulation and the application of a high dose of 400 U [12]. Another study mentions that the number of muscles treated and the dose per patient vary depending on the spasticity pattern, patient size, and the residual function of the affected limb [26]. A study in children with CP concluded that the efficacy of injections was not significantly better when higher doses of botulinum toxin were used [32]

### 4.3. Clinical Implications

Based on the evidence reviewed, botulinum toxin injections effectively decrease muscle tone in both limbs in poststroke spasticity, improving quality of life and showing to be a safe therapeutic agent [13,15]. In addition, botulinum toxin injections are effective in improving gait, range of motion, spasticity, and caregiver satisfaction in patients with CP, mainly when assessed in the medium- to short-term [19]. Despite the abundance of evidence on botulinum toxin injections, randomized clinical trials in different etiologies, adjusting for dosage, injection site, and age, are needed to adequately assess the efficacy of this therapy and generalize it to daily clinical practice.

### 4.4. Limitations

This study has several limitations. Firstly, several meta-analyses were conducted with a limited number of studies due to the low prevalence of disease spasticity-associated conditions. Secondly, there is variability in botulinum toxin doses and application sites among studies, which influences the results and complicates the generation of generalizable conclusions. Thirdly, several primary studies were included in more than one meta-analysis, potentially increasing the influence of these studies. Although these overlapping studies provided the most rigorous and consistent evidence, the possible presence of regression-to-mean bias prevents us from making a pooled estimate of the estimates from the analyses. Fourthly, the results presented in our umbrella review showed considerable heterogeneity, which may be due to differences between population groups, age ranges, types of botulinum toxin, and injection protocols, so these conclusions should be interpreted with caution.

## 5. Conclusions

In summary, the analyses in this umbrella review demonstrate the efficacy of botulinum toxin injections in reducing spasticity as measured by the MAS, both in patients with CP and poststroke, leading to an improvement in the patient’s quality of life. Botulinum toxin injection therapy is considered reversible and safe and can be used in conjunction with other treatments such as physiotherapy and physical activity. However, further randomized clinical trials conducted in populations with different etiologies and adjusted for dosage, injection site, and age are needed to generalize this therapy to daily clinical practice.

## Figures and Tables

**Figure 1 pharmaceuticals-17-00310-f001:**
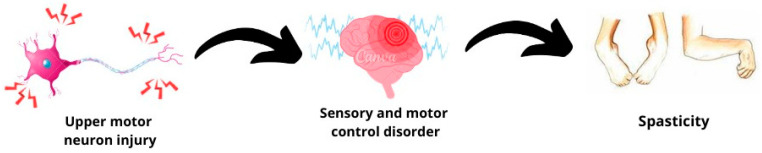
Spasticity definition.

**Figure 2 pharmaceuticals-17-00310-f002:**
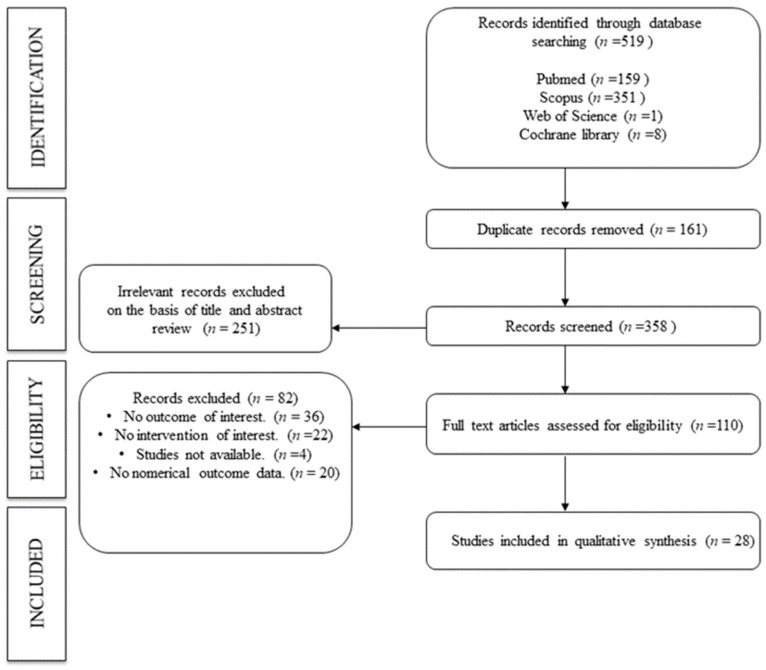
Flowchart. Search strategy.

**Figure 3 pharmaceuticals-17-00310-f003:**
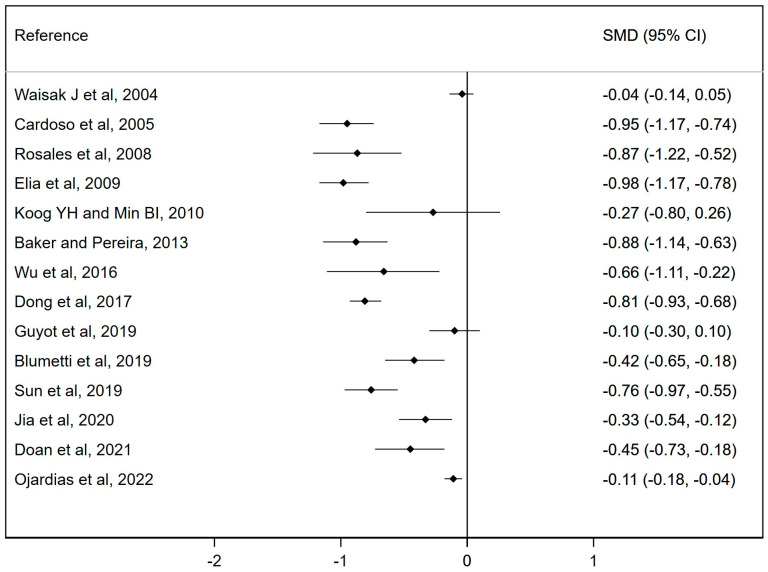
Forest plot for the effect of botulinum toxin injections on spasticity measured by the MAS [10,11,12,13,14,15,16,17,18,19,35,36,37,38].

**Figure 4 pharmaceuticals-17-00310-f004:**
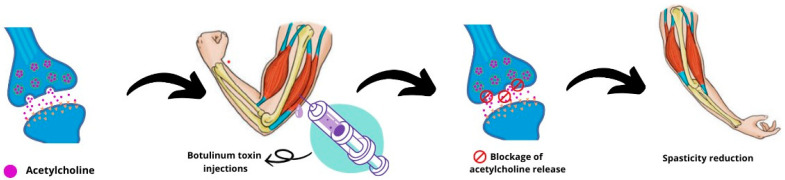
Mechanism of action of botulinum toxin injections on spasticity.

**Table 1 pharmaceuticals-17-00310-t001:** Characteristics of the included systematic reviews.

Reference	Studies Included (n)	Type of Population	Age Range (Years)	Intervention	Comparator	Length (Weeks)	Dossages U/kg/Muscle	Frequency	Injection Site	Spasticity Assessment Scale	Effect	Quality Assessment—AMSTAR 2
Reeuwijk, A et al. (2006) [22]	3 RCT and 9 US	Children with CP	1.0–18.0	BOTOX A	Placebo	NR	Botox: 75–300 U Dysport 500–1500 U	NR	Upper limb muscles	MAS	Insufficient evidence	Critically low
Demetrios, M. et al. (2013) [23]	3 RCT	Poststroke	NR	BOTOX A	Placebo	6–24	1000 units dysport	NR	Upper limb muscles	MAS	Low-level evidence	Moderate
Phadke, C.P. et al. (2014) [24]	5 CT	Poststroke	16.0–74.0	BOTOX	NR	4	NR	NR	Lower limb and upper limb muscles	MAS Tardieu Scale	Improvement	Moderate
García Salazar, L.F et al. (2015) [25]	17 Longitudinal Studies	Children with CP	2.0–22.0	BOTOX A	Placebo	2–24	NR	NR	Lower limb muscles	MAS Modified Tardieu Scale	Improvement	Moderate
Dashtipour, K. et al. (2015) [26]	12 RCT	Poststroke	NR	BOTOX A	Placebo	2–24	500–1500 U	Tree injections	Upper limb muscles	MAS	Safe and effective	Moderate
Fonseca Junior, P.R. et al. (2017) [27]	4 RCT	Children with CP	4.0–14.0	BOTOX A	NR	0–26	4–25.5 U/kg	Single dose	Lower limb muslces	Modified Tardieu Scale	Effective	Moderate
Gupta, A.D. et al. (2018) [28]	5 RCT	Poststroke	18.0–78.0	BOTOX A	Placebo	0–16	NR	NR	Lower limb muscles	MAS	Effective	Moderate
Yana, M. et al. (2019) [29]	4 RCT	Children with CP	2.0–8.0	BOTOX A	NR	8–48	100–500 U/muscle	6–8 injections	Lower limb muscles	MAS	Positive improvement	Moderate
Hara, T. et al. (2019) [30]	24 RCT and 2 comparative studies	Poststroke	41.2–67.0	BOTOX A	Placebo	1–27	500–1000 U	Multiple doses	Upper and lower limb muscles	MAS	Limited effectiveness	Moderate
Farag, S.M. et al. (2020) [31]	15 RCT	Children with CP	2.6–10.7	BOTOX A	Placebo	2–27	160–1000 U	Repeated sesions	Upper limb muscles	MAS Tardieu Scale	Improvement	Moderate
Klein, C. et al. (2023) [32]	24 RCT	Children with CP	3 y 1 mo–10 y	BOTOX A	Placebo	0–26	2–16 U/kg	NR	Upper limb muscles	MAS Tardieu Scale	Improvement	Moderate
Yang, H. et al. (2023) [33]	12 RCT	Children with CP	8 mo–10 y	BOTOX A	NR	0–24	0.5–20 U/kg	Single injection 6 months	Lower limb muscles	MAS Modified Tardieu Scale	Safety and efficacy	Moderate

CP: cerebral palsy; CT: clinical trial; MAS: Modified Ashworth Scale; mo: months; NR: not reported; RCT: randomized controlled trial; U: units; US: uncontrolled study.

**Table 2 pharmaceuticals-17-00310-t002:** Characteristics of the included meta-analyses.

Reference	Study Design	Studies Included (n)	Type of Population	Age Range (Years)	Intervention	Comparator	Length (Weeks)	Dosages	Frequency	Injection Site	Spasticity Assessment Scale	Effect SMD (95% CI)	Quality Assessment—AMSTAR 2
Boyd, R. and Hays, R. (2001) [34]	SR and meta-analysis	10 RCT	Children with CP	2.0–13.0	BOTOX A	Placebo	6–24	2–25.5 U/kg/muscle	4 weeks	Lower limb muscles	MAS	NR	Critically low
Wasiak, J. et al. (2004) [35]	SR and meta-analysis	2 RCT	Children with CP	2.5–10.0	BOTOX A	Placebo	2–12	2–9 U/kg/muscle	NR	Upper limb muscles	Tardieu Scale MAS	MAS −0.04 (−0.14, 0.05)	Low
Cardoso, E. et al. (2005) [16]	SR and meta-analysis	5 RCT	Poststroke	NR	BOTOX A	Placebo	4–6	NR	NR	Lower limb muscles	MAS	MAS −0.95 (−1.17, −0.74)	Low
Rosales, R. et al. (2008) [15]	SR and meta-analysis	9 RCT	Poststroke	NR	BOTOX A	Placebo	4–6	500–1500 U dysport 200–360 U Botox	NR	Upper limb muscles	MAS	MAS −0.87 (−1.22, −0.52)	Moderate
Elia, A.E. et al. (2009) [14]	SR and meta-analysis	11 RCT	Poststroke	NR	BOTOX A BOTOX B	Placebo	3–6 9–12	10–3750 U/muscle	Regular intervals	Upper limb muscles	MAS	MAS −0.98 (−1.17, −0.78)	High
Koog, Y.H. and Min, B.I. (2010) [36]	SR and meta-analysis	15 RCT	Children with CP	<12.0	Botox, HengLi, and Dysport	NR	4–16	2.8–30 U/kg	NR	Lower limb muscles	MAS		High
Baker, J.A. and Pereira, G. (2013) [13]	SR and meta-analysis	37 RCT	Adults with spasticity	NR	BOTOX A	Placebo	4–12	NR	Single dose	Upper limb and lower limb muscles	MAS Modified Tardieu Scale	MAS −0.88 (−1.14, −0.63)	Moderate
Wu, T. et al. (2016) [12]	SR and meta-analysis	7 RCT	Poststroke	14.0–85.0	BOTOX	Placebo	8–24	100–400 U	NR	Lower limb muscles	MAS	MAS −0.66 (−1.11, −0.22)	High
Dong, Y. et al. (2017) [11]	SR and meta-analysis	22 RCT	Poststroke	57.6	BOTOX A	Placebo	2–24	80–1500 U	NR	Lower limb muscles	MAS Tardieu Scale	MAS −0.81 (−0.93, −0.68)	High
Guyot, P. et al. (2019) [18]	SR and network meta-analysis	10 RCT	Children with spasticity	1.7–7.4	BOTOX A	Placebo	4–12	0.5–30 U/kg/leg	NR	Lower limb muscles	MAS Tardieu Scale	MAS −0.1 (−0.3, 0.1)	High
Blumetti, F.C. et al. (2019) [19]	SR and meta-analysis	31 RCT in SR 28 RCT in meta-analysis	Children with CP	1.3–9.5	BOTOX A	Placebo	2–48	4–30 U/kg	3–6 months 2–4 weeks	Lower limb muscles	MAS Modified Tardieu Scale	Modified Tardiue Scale. −0.83 (−0.98, −0.67) MAS −0.42 (−0.65, −0.18)	High
Sun, L.C. et al. (2019) [10]	SR and meta-analysis	27 RCT	Poststroke	49.3–63.5	BOTOX A	Placebo	4–24	75–1500 U	NR	Upper limb and lower limb muscles	MAS	MAS −0.76 (−0.97, −0.55)	High
Jia, S. et al. (2020) [37]	SR and meta-analysis	10 RCT	Poststroke	18.0–92.0	BOTOX A	Placebo	12–24	75–1500 U	NR	Upper limb muscles	MAS	MAS −0.33 (−0.54, −0.12)	High
Doan, T.N. et al. (2021) [38]	SR and meta-analysis	12 RCT	Poststroke	NR	BOTOX A	Placebo	NR	Botox: 100–540 U Dysport: 500–1500 U	NR	Lower limb muscles	MAS	MAS −0.45 (−0.73, −0.18)	High
Varvarousis, D.N. et al. (2021) [8]	SR and meta-analysis	21 CCT and RCT	Poststroke	NR	BOTOX A	NR	NR	<200 U, >200 U	NR	Upper limb and lower limb muscles	MAS	NR	High
Ojardias, E. et al. (2022) [17]	SR and meta-analysis	37 RCT	Poststroke	52.0–67.0	BOTOX A	Placebo	NR	NR	Single injection 00	Upper limb muscles	MAS	MAS −0.11 (−0.18, −0.04)	High

CCT: clinical controlled trial; CI 95%: 95% confidence interval; CP: cerebral palsy; MAS: Modified Ashworth Scale; NR: not reported; RCT: randomized controlled trial; SMD: standard mean difference; U: units.

## Data Availability

Not applicable.

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
