# Peer review of "Effect of Botulinum Toxin Injections in the Treatment of Spasticity of Different Etiologies: An Umbrella Review"

_pharmaceuticals, 2024, doi:10.3390/ph17030310_

Round 1

Reviewer 1 Report

Comments and Suggestions for Authors

The review includes studies up to January 2024. However, it should be verified whether any relevant studies were published just after this date but before the manuscript's submission, as this could impact the review's comprehensiveness.

While the paper outlines the use of standardized mean differences (SMDs) for data synthesis, a more detailed explanation of the statistical methods used to handle heterogeneity among studies (e.g., random vs. fixed effects models) would enhance the readers' understanding of the analysis.

The paper assesses the methodological quality of included studies using the AMSTAR 2 tool. It would be beneficial to provide more detailed findings of this assessment, perhaps in a supplementary material, to give readers insight into the quality and reliability of the evidence base.

The paper reports considerable heterogeneity in the meta-analysis results. A more thorough discussion regarding the sources of this heterogeneity (e.g., differences in study populations, botulinum toxin types, and injection protocols) and its implications for the interpretation of the findings would be valuable.

Author Response

REVIEWER 1

The review includes studies up to January 2024. However, it should be verified whether any relevant studies were published just after this date but before the manuscript's submission, as this could impact the review's comprehensiveness.

Authors

Thank you for the reviewer's comment. We greatly appreciate the reviewer´s time in reviewing the manuscript. As suggested, we have performed the search again in the different databases to include possible potential studies, but no new studies were found according to our search strategy. We have modified the search date in the abstract and in the methods section.

“Methods: A systematic search of different databases, including Pubmed, Scopus, the Cochrane Library, and Web of Science, was performed from inception to February 2024. […]”

“A search was conducted in the Pubmed, Scopus, Cochrane, and Web of Science databases from their inception to February 2024. […]”

While the paper outlines the use of standardized mean differences (SMDs) for data synthesis, a more detailed explanation of the statistical methods used to handle heterogeneity among studies (e.g., random vs. fixed effects models) would enhance the readers' understanding of the analysis. 

Authors:

Thank you for the reviewer’s comment. As suggested, we have modified this issue as follows:

The DerSimonian and Laird random-effects method [24] was used to calculate pooled estimates of standardized mean differences (SMDs) and their respective 95% confidence intervals (95% CIs) to assess the effect of botulinum toxin injections com-pared to that of a control group using the MAS scale and are displayed in a forest plot.”

In the references section:

  1. DerSimonian, R.; Kacker, R. Random-effects model for meta-analysis of clinical trials: an update. Contemp Clin Trials. 2007; 28: 105-14

The paper assesses the methodological quality of included studies using the AMSTAR 2 tool. It would be beneficial to provide more detailed findings of this assessment, perhaps in a supplementary material, to give readers insight into the quality and reliability of the evidence base. 

Authors:

Thank you for the reviewer's comment. As suggested, we have included the following information:

In the Methods section:

“[…] The overall quality of the studies was rated as high when there are no deficiencies or only one non-critical deficiency; moderate when there is more than one non-critical deficiency; low when there is a critical deficiency with or without non-critical deficiencies; and critically low with more than one critical deficiency with or without non-critical deficiencies.”

In the Supplementary material:

“[…] 1) Did the research questions and inclusion criteria for the review include the components of PICO? 2) Did the report of the review contain an explicit statement that the review methods were established prior to the conduct of the review and did the report justify any significant deviations from the protocol? 3) Did the review authors explain their selection of the study designs for inclusion in the review? 4) Did the review authors use a comprehensive literature search strategy? 5) Did the review authors perform study selection in duplicate? 6) Did the review authors perform data extraction in duplicate? 7) Did the review authors provide a list of excluded studies and justify the exclusions? 8) Did the review authors describe the included studies in adequate detail? 9) Did the review authors use a satisfactory technique for assessing the risk of bias (RoB) in individual studies that were included in the review? 10) Did the review authors report on the sources of funding for the studies included in the review? 11) If meta-analysis was performed did the review authors use appropriate methods for statistical combination of results? 12) If meta-analysis was performed, did the review authors assess the potential impact of RoB in individual studies on the results of the meta-analysis or other evidence synthesis? 13) Did the review authors account for RoB in individual studies when interpreting/ discussing the results of the review? 14) Did the review authors provide a satisfactory explanation for, and discussion of, any heterogeneity observed in the results of the review? 15) If they performed quantitative synthesis did the review authors carry out an adequate investigation of publication bias (small study bias) and discuss its likely impact on the results of the review? 16) Did the review authors report any potential sources of conflict of interest, including any funding they received for conducting the review?

The paper reports considerable heterogeneity in the meta-analysis results. A more thorough discussion regarding the sources of this heterogeneity (e.g., differences in study populations, botulinum toxin types, and injection protocols) and its implications for the interpretation of the findings would be valuable. 

Authors:

The reviewer´s comment seems judicious. As suggested, we have modified this issue as follows:

“[…] A study in post-stroke patients, in which different doses were analyzed, reports that regardless of the dose and type of botulinum toxin used, there was a reduction in the MAS scale, with the maximum reduction occurring between weeks 4 and 6 post-injection [16]. One of the reviewed studies reports that there was no difference in spasticity reduction between the application of 100 U of botulinum toxin combined with short-term electrical stimulation and the application of a high dose of 400 U [12]. Another study mentions that the number of muscles treated and the dose per patient vary depending on the spasticity pattern, patient size, and the residual function of the affected limb [32]. A study in children with CP concluded that the efficacy of injections was not significantly better when higher doses of botulinum toxin were used [28]. […]”

“[…] Fourth, the results presented in our umbrella review showed considerable heterogeneity, which may be due to differences between population groups, age ranges, types of botulinum toxin and injection protocols, so these conclusions should be interpreted with caution. […]”

Reviewer 2 Report

Comments and Suggestions for Authors

Overall, the article provides a comprehensive overview of the effect of botulinum toxin injections in treating spasticity. However, there are a few potential concerns that should be addressed:

1-     While the article mentions that a systematic search was conducted in various databases, it does not provide detailed information about the search strategy employed. The inclusion and exclusion criteria are mentioned briefly, but additional information on the keywords used and the search process would enhance the transparency and reproducibility of the study.

2-     Excluding articles based on language in this ambarella review study can introduce language bias, limit generalizability, miss relevant studies, reduce the diversity of evidence, and overlook valuable contributions from non-English/non-Spanish publications. I am curious to know if the authors had carefully weighed these disadvantages against the available resources and the specific research question and objectives of this review.

3-     Studies of diverse populations (i.e., pediatric, and adult patients) have been included. The variations in age can lead to differences in treatment responses and outcomes. Also, this heterogeneity may limit the ability to draw firm conclusions or conduct meaningful meta-analyses. Address whether any subgroup analyses were conducted to explore the influence of this factor and its implications for clinical practice.

4-     Although the article states that 28 studies were included in the umbrella review, it does not provide a full summary or characteristics of these studies. I would suggest providing additional details regarding treatment protocols (dosages, frequency, and injection sites) in Table 1. This would allow readers to assess the relevance and generalizability of the findings.

5-     Include a section discussing the clinical implications of the findings and explain how the results of the review can inform clinical practice, guidelines, or future research directions. This will enhance the practical relevance of the study.

6-     Also, incorporating a discussion on the safety and adverse effects of botulinum toxin injections in the treatment of spasticity would be interesting. Discuss any reported adverse events from the included studies and their potential impact on patient outcomes.

7-     And more, outlining the potential mechanisms of action underlying the therapeutic effects of botulinum toxin injections for spasticity could enhance the understanding of how it works at a physiological level and its implications for optimizing patient outcomes

8-     The article briefly mentions previous systematic reviews and meta-analyses on the topic, but it does not discuss potential limitations or biases in these studies. So, it would be valuable to provide a critical appraisal of the existing evidence, including the quality of the included studies, heterogeneity of the findings, and potential sources of bias.

9-     A clear conclusion would help readers understand the key takeaways from the study and its potential impact on clinical practice or future research directions.

Author Response

REVIEWER 2

Overall, the article provides a comprehensive overview of the effect of botulinum toxin injections in treating spasticity. However, there are a few potential concerns that should be addressed:

Authors:

Thank you for the reviewer’s comment. We greatly appreciate the reviewer´s time in reviewing the manuscript.

1- While the article mentions that a systematic search was conducted in various databases, it does not provide detailed information about the search strategy employed. The inclusion and exclusion criteria are mentioned briefly, but additional information on the keywords used and the search process would enhance the transparency and reproducibility of the study.

Authors:

Thank you for the reviewer’s comment. As suggested, we have modified this issue as follows:

“[…] The selected keywords were combined using Boolean operators (AND, OR) following the population, intervention, comparator, outcomes (PICO) strategy to identify studies assessing the effect of botulinum toxin in the treatment of spasticity (Supplementary Table 1). The following search strategy was used: ((“general population”) OR (“children”) OR (“adults”)) AND ((“botox”) OR (“botulinum toxin”)) AND ((“spasticity”) OR (“cerebral palsy”) OR (“spastic paraplegia)) AND ((“systematic review”) OR (“meta-analysis”) OR (“network meta-analysis”)). […]”

2- Excluding articles based on language in this ambarella review study can introduce language bias, limit generalizability, miss relevant studies, reduce the diversity of evidence, and overlook valuable contributions from non-English/non-Spanish publications. I am curious to know if the authors had carefully weighed these disadvantages against the available resources and the specific research question and objectives of this review.

Authors:

The reviewer´s comment seems judicious. As suggested, we have searched again without language restriction, and have found no new studies that meet the inclusion criteria. In addition, we have modified the inclusion/exclusion criteria in the manuscript. 

“[…] The inclusion criteria were as follows: i) population: general population (children and adults); ii) intervention: botulinum toxin injections; iii) outcome: spasticity assessed with the MAS and/or Tardieu scale; and iv) study design: systematic reviews and/or meta-analyses; v) no language restriction. […]”

3- Studies of diverse populations (i.e., pediatric, and adult patients) have been included. The variations in age can lead to differences in treatment responses and outcomes. Also, this heterogeneity may limit the ability to draw firm conclusions or conduct meaningful meta-analyses. Address whether any subgroup analyses were conducted to explore the influence of this factor and its implications for clinical practice.

Authors:

Thank you for the reviewer's comment. As suggested, we have included this issue as a limitation of the study because, being a review of reviews, it is not possible to perform subgroup analysis as each meta-analysis included has different types of populations.

“[…] Fourth, the results presented in our umbrella review showed considerable heterogeneity, which may be due to differences between population groups, age ranges, types of botulinum toxin and injection protocols, so these conclusions should be interpreted with caution. […]”

4- Although the article states that 28 studies were included in the umbrella review, it does not provide a full summary or characteristics of these studies. I would suggest providing additional details regarding treatment protocols (dosages, frequency, and injection sites) in Table 1. This would allow readers to assess the relevance and generalizability of the findings. 

Authors

The reviewer´s comment seems judicious. As suggested, we have included dosages, frequency, and injection sites in Tables 1 and 2.

5- Include a section discussing the clinical implications of the findings and explain how the results of the review can inform clinical practice, guidelines, or future research directions. This will enhance the practical relevance of the study. 

Authors:

Thank you for the reviewer’s comment. As suggested, we have included a section discussing the clinical implications of the findings and explain how the results of the review can inform clinical practice, guidelines, or future research directions.

Based on the evidence reviewed, botulinum toxin injections effectively decrease muscle tone in both limbs in post-stroke spasticity, improving quality of life and showing to be a safe therapeutic agent [13, 15]. In addition, botulinum toxin injections are effective in improving gait, range of motion, spasticity, and caregiver satisfaction in patients with CP, mainly when assessed in the medium to short term [19]. Despite the abundance of evidence on botulinum toxin injections, randomized clinical trials in different aetiologies, adjusting for dosage, injection site, and age are needed to adequately assess the efficacy of this therapy and generalize it to daily clinical practice.”

6- Also, incorporating a discussion on the safety and adverse effects of botulinum toxin injections in the treatment of spasticity would be interesting. Discuss any reported adverse events from the included studies and their potential impact on patient outcomes.

Authors

Thank you for the reviewer's comment. As suggested, we have included this issue as follows:

“[…] Based on the studies reviewed, this therapy is generally well tolerated and safe, although botulinum toxin injections can cause fatigue, tiredness, pain, skin rashes, flu-like symptoms, worsening of spasms, weakness, convulsions, and incoordination [12, 32]. Compared to other therapies, such as intrathecal baclofen, the incidence of treatment-related adverse effects is low [33]

7- And more, outlining the potential mechanisms of action underlying the therapeutic effects of botulinum toxin injections for spasticity could enhance the understanding of how it works at a physiological level and its implications for optimizing patient outcomes. 

Authors

Thank you for the reviewer's comments. As suggested, we have included the mechanism of action of botulinum toxin injections for spasticity. In addition, we have included a figure to better explain the mechanism of action of botulinum toxin injections on spasticity.

“[…] Neuromuscular blockade affects both intrafusal and extrafusal muscle fibers. The decrease in activity of intrafusal muscle fibers (reduction in afferent input) peaks at 2 weeks and gradually decreases by 12 weeks post-injection. This blockade reduces the flow of muscle spindles to spinal stretch reflex circuits, reducing spasticity [2]. (Figure 3) […]”.

8- The article briefly mentions previous systematic reviews and meta-analyses on the topic, but it does not discuss potential limitations or biases in these studies. So, it would be valuable to provide a critical appraisal of the existing evidence, including the quality of the included studies, heterogeneity of the findings, and potential sources of bias. 

Authors:

Thank you for the reviewer's comment. As suggested, we have assessed these limitations using the GRADE tool.

In the Methods section:

“[…] We used the Grading of Recommendations, Assessment, Development, and Evaluation (GRADE) tool [13] to assess the quality of evidence and provide recommendations. Each outcome had a high, moderate, low, or very low evidence score, depending on the study design, risk of bias, inconsistency, indirect evidence, imprecision, and publication bias. […]”

In the Results section:

“[…] When the GRADE was evaluated, 92.86 of the pairwise comparison studies were rated as low and 7.14 as very low (Supplementary Table 6). […]”

In the References section:

  1. Guyatt GH, Oxman AD, Schünemann HJ, Tugwell PKnottnerus A. GRADE guidelines: A new series of articles in the Journal of Clinical Epidemiology. J Clin Epidemiol. 2011;64:380-382.

In supplementary material: (see the following table)

9. A clear conclusion would help readers understand the key takeaways from the study and its potential impact on clinical practice or future research directions.

Authors:

The reviewer’s comment seems judicious. As suggested, we have included a clearer conclusion.

In summary, the analyses in this umbrella review demonstrate the efficacy of botulinum toxin injections in reducing spasticity as measured by the MAS scale, both in patients with CP and post-stroke, leading to an improvement in the patient's quality of life. Botulinum toxin injections therapy is considered reversible and safe, and can be used in conjunction with other treatments such as physiotherapy and physical activity. However, further randomized clinical trials conducted in populations with different aetiologies and adjusted for dosage, injection site, and age are needed to generalize this therapy to daily clinical practice.”

Reviewer 3 Report

Comments and Suggestions for Authors

The overall composition of the manuscript is good. The paper is scientifically and methodologically accurate. This manuscript will interest many readers.

However, my recommendation is 'Minor Revision'. More detailed comments are given below.

General comments:

1. Although the manuscript is submitted as a review, it contains methods, results, and discussion sections. Given the above, I believe this manuscript is not a review paper and should be submitted as a research article with detailed experimental procedures, enabling reviewers to critically assess the data and conclusions.

1. The main problem is that the authors must justify what is the advantage of their study with respect to previous studies, and what are the new contributions of this manuscript with respect to others.

Introduction section

3. For better understanding of readers.  Two scales used specifically to assess spasticity. The first is the Modified Ashworth Scale (MAS) and the second is the Tardieu Scale or the Modified Tardieu Scale. A diagram of the two scales must be provided.

4. The authors need to reinforce their focus on why they carry out this work. The way as presented is weak. Before the Materials and Methods section, the authors must indicate which strategies were used to achieve their objectives.

Results

5. A short introduction and conclusion in each section must be provided before describing all the experiments and results that support each section. It is to guide the reader to understand the importance of the results.

6. Table 1 and 2. The references are not provided according to guidelines to Pharmaceutical journal. The asme for Figure 2. Please modify.

7. Although, there are a substantial volume of papers concerning to effectiveness of botulinum toxin injections compared to the control group research, is not clear the associations between academic journals and effectiveness of botulinum toxin injections. I would like the authors to relate their results obtained in this study with the results obtained in the therapy. This would substantially improve the work.This should be improved.

Discussion section

8. Lines 169-171: The authos menion  that “This study provided significant evidence on the effectiveness of Botulinum toxin injections in reducing muscle tone, as measured by the MAS, as indicated by a decrease in spasticity”. However, in the reslts section nothing is mentioned regarding this point.

9. A schematic must be provided that relate the information provided in the discussion section.

10. For a better understanding of readers. It is suggested that authors include figures in their main document. For example, a figure must be included in the introduction and discussion sections. Please add this information

11. The Discussion  section must be improved. In the current format it is limited.

Author Response

REVIEWER 3

The overall composition of the manuscript is good. The paper is scientifically and methodologically accurate. This manuscript will interest many readers.

However, my recommendation is 'Minor Revision'. More detailed comments are given below.

Authors

Thank you for the reviewer’s comment. We greatly appreciate the reviewer´s time in reviewing the manuscript.

General comments:

1. Although the manuscript is submitted as a review, it contains methods, results, and discussion sections. Given the above, I believe this manuscript is not a review paper and should be submitted as a research article with detailed experimental procedures, enabling reviewers to critically assess the data and conclusions.

Authors:

Thank you for the reviewer’s comment. As suggested, we have changed the category of the manuscript from “review” to “research article”.

2. The main problem is that the authors must justify what is the advantage of their study with respect to previous studies, and what are the new contributions of this manuscript with respect to others. 

Authors.

The reviewer´s comment seems judicious. As suggested, we have justified this issue as follows:

“[…] Since no study has synthesized all the existing information on the effect of botulinum toxin injections, an umbrella review is needed to synthesize this information and evaluate the effect of the treatment on the reduction of spasticity. […]”

Introduction section

3. For better understanding of readers. Two scales used specifically to assess spasticity. The first is the Modified Ashworth Scale (MAS) and the second is the Tardieu Scale or the Modified Tardieu Scale. A diagram of the two scales must be provided.

Authors

Thank you for the reviewer's comment. As suggested, we have included two tables to the supplementary material, one for the Modified Ashworth Scale and one for the Modified Tardieu Scale. 

4. The authors need to reinforce their focus on why they carry out this work. The way as presented is weak. Before the Materials and Methods section, the authors must indicate which strategies were used to achieve their objectives.

Authors:

Thank you for the reviewer’s comment. As suggested, we have included this issue as follows: 

“[…] Since no study has synthesized all the existing information on the effect of botulinum toxin injections, an umbrella review is needed to synthesize this information and evaluate the effect of the treatment on the reduction of spasticity. […]”

Results

5. A short introduction and conclusion in each section must be provided before describing all the experiments and results that support each section. It is to guide the reader to understand the importance of the results.

Authors:

The reviewer’s comment seems judicious. As suggested, we have added a short introduction and conclusion to each results section.

  • Baseline characteristics

“[…] The flowchart was created to summarize how the data extraction process was carried out in this manuscript. […]”

“[…] In summary, two tables were created, one for systematic reviews and one for meta-analysis, which synthesize the data obtained from the records included in this study. […]”

  • Methodological quality assessment

“[…] To assess the quality of the studies, it is necessary to use a specific scale for systematic reviews and meta-analysis. […]”

“[…] In summary, most of the studies have achieved moderate quality in AMSTAR 2 tool and most of the studies were rated as low in GRADE tool. […]”

  • Data Synthesis

“[…] A forest plot of the included meta-analyses has been conducted to assess the effectiveness of botulinum toxin injections on spasticity.

“[…] In summary, in all studies, the results were favourable to the intervention, showing significant results in in 78.57% of the studies. […]”

6. Table 1 and 2. The references are not provided according to guidelines to Pharmaceutical journal. The same for Figure 2. Please modify.

Authors

Thank you for the reviewer´s comment. As suggested, we modified the corresponding refences in Table 1, Table 2, and Figure 2.

7. Although, there are a substantial volume of papers concerning to effectiveness of botulinum toxin injections compared to the control group research, is not clear the associations between academic journals and effectiveness of botulinum toxin injections. I would like the authors to relate their results obtained in this study with the results obtained in the therapy. This would substantially improve the work. This should be improved.

Authors:

Thank you for the reviewer’s comment. As suggested, we have included this issue in the results section comparing our findings with those of clinical practice.

“3.4. Effectiveness of botulinum toxin injection

The impact of botulinum toxin injections in clinical practice needs to be evaluated, as shown below: Our results indicate that botulinum toxin decreases spasticity assessed by the MAS scale. Studies suggest that in clinical practice, this therapy reduces spasticity, decreasing muscle tone, resulting in a reduction in the MAS scale [15,19]. This reduction occurs both in post-stroke patients [15] and in children with CP [19].

In summary, both in the results obtained and in clinical practice, a reduction in spasticity is showed.”

Discussion section

8. Lines 169-171: The authors mention that “This study provided significant evidence on the effectiveness of Botulinum toxin injections in reducing muscle tone, as measured by the MAS, as indicated by a decrease in spasticity”. However, in the results section nothing is mentioned regarding this point.

Authors

Thank you for the reviewer´s comment. As suggested, we added the following issue in Data Synthesis.

“[…] Therefore, there is a significant reduction in spasticity and a reduction of muscle tone according to the MAS scale. […]”

9. A schematic must be provided that relate the information provided in the discussion section.

Authors:

The reviewers comment seems judicious. As suggested, we have structured the discussion section as follows:

“4.1. Main findings”

“4.2. Interpretation”

“4.3. Clinical implications”

“4.4. Limitations”

10. For a better understanding of readers. It is suggested that authors include figures in their main document. For example, a figure must be included in the introduction and discussion sections. Please add this information. 

Authors

Thank you for the reviewer's comment. As suggested, we have included figures in the introduction and in the discussion sections.

“Figure 1: Spasticity definition.”

“Figure 3: Mechanism of action of botulinum toxin injections on spasticity”

11. The Discussion section must be improved. In the current format it is limited.

Authors

The reviewer´s comment seems judicious. As suggested, we have modified the discussion section as follows: 

“4.1 Main findings”

“4.2 Interpretation”

“[…] Botulinum toxin, when injected at high concentrations, strongly affects the management of spasticity by blocking the release of the neurotransmitter acetylcholine from the neuromuscular junction to various muscle groups, resulting in what is known as "chemical denervation". Neuromuscular blockade affects both intrafusal and extrafusal muscle fibers. The decrease in activity of intrafusal muscle fibers (reduction in afferent input) peaks at 2 weeks and gradually decreases by 12 weeks post-injection. This blockade reduces the flow of muscle spindles to spinal stretch reflex circuits, reducing spasticity [2]. (Figure 3) […]”

“[…] Based on the studies reviewed, this therapy is generally well tolerated and safe, although botulinum toxin injections can cause fatigue, tiredness, pain, skin rashes, flu-like symptoms, worsening of spasms, weakness, convulsions, and incoordination [12, 32]. Compared to other therapies, such as intrathecal baclofen, the incidence of treatment related adverse effects is low [33]. […]”

“[…] A study in post-stroke patients, in which different doses were analyzed, reports that regardless of the dose and type of botulinum toxin used, there was a reduction in the MAS scale, with the maximum reduction occurring between weeks 4 and 6 post-injection [16]. One of the reviewed studies reports that there was no difference in spasticity reduction between the application of 100 U of botulinum toxin combined with short-term electrical stimulation and the application of a high dose of 400 U [12]. Another study mentions that the number of muscles treated and the dose per patient vary depending on the spasticity pattern, patient size, and the residual function of the affected limb [32]. A study in children with CP concluded that the efficacy of injections was not significantly better when higher doses of botulinum toxin were used [28]. […]”

“4.3 Clinical implications”

“[…] Based on the evidence reviewed, botulinum toxin injections effectively decrease muscle tone in both limbs in post-stroke spasticity, improving quality of life and showing to be a safe therapeutic agent [13, 15]. In addition, botulinum toxin injections are effective in improving gait, range of motion, spasticity, and caregiver satisfaction in patients with CP, mainly when assessed in the medium or short term [19]. Despite the abundance of evidence on botulinum toxin injections, randomized clinical trials in different aetiologies, adjusting for dosage, injection site, and age are needed adequately assess the efficacy of this therapy and to generalize it to daily clinical practice. […]”

“4.4 Limitations”

“[…] Fourth, the results presented in our umbrella review showed considerable heterogeneity, which may be due to differences between population groups, age ranges, types of botulinum toxin and injection protocols, so these conclusions should be interpreted with caution. […]”

Round 2

Reviewer 2 Report

Comments and Suggestions for Authors

The authors have incorporated the revisions suggested in the previous review round. They have responded to the comments, clarifying their methodology, addressing potential limitations, and strengthening their arguments. The revisions have significantly enhanced the overall coherence and readability of the manuscript. I have no further comments.